# Optimization of acquisition time and reconstruction parameters for quantitative single-photon emission computed tomography/computed tomography using iodine-123 metaiodobenzylguanidine

**Masakazu Tsujimoto**[1]*, **Masanori Watanabe**[2], **Kenta Nogami**[2], **Hideki Kawai**[3], **Masayoshi Sarai**[3]

**1** Department of Medical Equipment Engineering, Clinical Collaboration Unit, School of Medical Sciences, Fujita Health University, Toyoake, Japan, **2** Department of Radiology, Fujita Health University Hospital, Toyoake, Japan, **3** Department of Cardiology, Fujita Health University Hospital, Toyoake, Japan

* mckz-t@fujita-hu.ac.jp

## Abstract

This study aimed to determine the optimal measurement conditions for accurate standardized uptake value (SUV) analysis of iodine-123 metaiodobenzylguanidine ($^{123}$I-MIBG) by examining the relationship between image convergence and quantitation. Single-photon emission computed tomography/computed tomography images were acquired using JS-10 and National Electrical Manufacturers Association (NEMA) body phantoms, with acquisition time per view varied (10, 30, 50, and 100 s/view). Image reconstruction was performed using three-dimensional-ordered subset expectation maximization, adjusting the product of subset and iteration (SI product; 60, 120, 180) and Gaussian filter parameters (8, 10, 12 mm). For the JS-10 phantom, we evaluated the dose linearity (DL), the recovery coefficient (RC) of individual rods, the scatter ratio (SR), and the coefficient of variation (CV). For the NEMA body phantom, we assessed the contrast-to-noise ratio (CNR) of the 17-mm-diameter hot sphere. We also evaluated the maximum and mean SUVs for all its hot spheres, and their relative standard error (RSE), using SUVs obtained at 100 s/view as reference. In the JS-10 phantom, the DL remained stable under all conditions. The RC decreased when the Gaussian filter was large and the SI product was small. A trade-off between the CV and the SR emerged, depending on the acquisition time and the SI product; optimal results were observed at 50 – 100 s/view and an SI product of 120 – 180. In the NEMA body phantom, contrast improved with acquisition times of ≥30 s/view, and the CNR increased as noise declined with longer acquisition times. At ≥50 s/view, variation in the maximum and mean SUVs decreased, with the RSE remaining below 5%. In conclusion, accurate SUV measurement with $^{123}$I-MIBG requires an acquisition time of ≥50 s/view, an SI product of approximately 120, and a

**Data availability statement:** All relevant data are within the paper and its Supporting Information files.

**Funding:** MS received partial financial support from PDRadiopharma Inc. The funders had no role in study design, data collection and analysis, decision to publish, or preparation of the manuscript.

**Competing interests:** The authors have declared that no competing interests exist.

Gaussian filter of 10 − 12 mm. These findings provide a foundation for future studies comparing this method with the heart-to-mediastinum ratio, supporting its clinical application.

## Introduction

Myocardial scintigraphy using iodine-123 metaiodobenzylguanidine ([123]I-MIBG) is a well-established technique widely used to assess cardiac sympathetic denervation [1,2]. Because [123]I-MIBG is structurally similar to norepinephrine, it is taken up by cardiac myocytes and allows visualization of sympathetic nerve activity in the heart. This method can sensitively detect functional abnormalities in the adrenergic nervous system in patients with cardiac conditions such as heart failure. Therefore, it plays a key role in understanding disease pathology and assessing treatment effectiveness [3]. Reduced cardiac uptake due to sympathetic denervation has also been reported in patients with neurodegenerative disorders, including Parkinson's disease and dementia with Lewy bodies. Accordingly, [123]I-MIBG imaging is expected to serve as a valuable diagnostic tool for differentiating among neurodegenerative diseases [4].

Conventionally, the semiquantitative index known as the heart-to-mediastinum (H/M) ratio—calculated by placing a region of interest (ROI) over the myocardium and mediastinum on planar images—is widely used to evaluate [123]I-MIBG uptake [5–7]. The H/M ratio is easy to measure; however, it is limited by overlapping signals from surrounding tissues, such as the liver, lungs, and blood pool, which complicate accurate assessment of regional myocardial function. Single-photon emission computed tomography (SPECT) is useful for evaluating local myocardial characteristics [8]. Three-dimensional assessment using SPECT improves accuracy by eliminating tissue overlap artifacts inherent to planar imaging. Commercial software capable of calculating standardized uptake values (SUVs) from SPECT images is increasingly available and is already in clinical use for bone SPECT imaging [9]. However, the clinical significance of the SUV quantification with [123]I-MIBG remains unclear. Basic studies on quantitative analysis exist [10,11]; however, they primarily address the accuracy of evaluating tracer accumulation in localized regions such as the myocardium and tumors. In studies of cardiac sarcoidosis using fluorodeoxyglucose-positron emission tomography, researchers have investigated the relationship between the myocardium and surrounding tissues [12,13]. These studies have shown that using SUV ratios with the aorta as a reference improves accuracy in detecting high-uptake areas compared with evaluating the myocardium alone. However, with [123]I-MIBG SPECT, foundational validation of the quantitative relationship between the myocardium and surrounding tissues remains incomplete. Further research, including foundational studies to define optimal acquisition and reconstruction parameters for SUV analysis, is essential to support the routine clinical use of [123]I-MIBG SPECT.

Therefore, in this study, we aimed to establish evidence for accurately quantifying SUVs in [123]I-MIBG SPECT/CT, particularly in cases of reduced myocardial uptake

such as cognitive disorders. To achieve this, we conducted a foundational investigation of the relationship between imaging parameters and quantitative accuracy for SUV measurement.

## Materials and methods

### Equipment and phantoms

The SPECT/CT scanner used was a Symbia T6 (Siemens K. K., Tokyo, Japan) equipped with a low-to-medium-energy general-purpose collimator. Image reconstruction was performed using the Syngo MI Application VB10B. The quantitative accuracy of SPECT imaging is limited, primarily due to its poorer spatial resolution compared to positron emission tomography images and the sensitivity of three-dimensional ordered subset expectation maximization (3D-OSEM) reconstruction algorithm to acquisition and reconstruction parameters. To establish optimal conditions for quantitative analysis, we used phantoms containing rod and spherical structures, as described in previous studies [14,15]. Specifically, we used the SPECT performance evaluation phantom JS-10 (Kyoto Kagaku Co., Ltd., Kyoto, Japan) and the National Electrical Manufacturers Association (NEMA) International Electrotechnical Commission (IEC) Body phantom (AcroBio Corp., Tokyo, Japan). These phantoms were used to evaluate structural detail under varying object configurations. We performed image analysis using ImageJ software (National Institutes of Health, MD, USA) and RAVAT ver1.2 (Nihon Medi-Physics Co., Ltd., Tokyo, Japan).

For phantom preparation, we first measured a stock solution of $^{123}$I using a calibrated dose calibrator. To prepare the rod sections of the JS-10 phantom and the spheres of the NEMA IEC body phantom, we diluted the stock solution to achieve predetermined target activity concentrations, accounting for radioactive decay up to the planned imaging start time. For the background compartments of both phantoms, we added a calculated volume of the stock solution—based on dose calibrator measurements—directly to the compartments. Thereafter, the phantoms were filled to their known volumes with water and mixed thoroughly to ensure uniform activity distribution.

Prior to all phantom imaging, the established system calibration procedures were followed. The dose calibrator used to measure the radioactivity of $^{123}$I was cross-calibrated with a well counter. To verify the accuracy of the final in-phantom activity concentrations, we collected 1 mL aliquots from each prepared solution—for both the target inserts and background compartments of the JS-10 and NEMA body phantoms—and measured them using the cross-calibrated well counter immediately before and after phantom filling. Furthermore, a SPECT system calibration factor (e.g., Becquerel calibration factor, BCF) to convert image measured count data into activity concentration (Bq/mL) was established by imaging a cylindrical phantom (inner diameter: 16 cm, height: 15 cm, total volume: 3,016 mL) with a known radioactivity concentration of 123I, also determined by the calibrated dose calibrator.

### SPECT/CT imaging

SPECT acquisition parameters were set according to a standard protocol tailored to myocardial thickness. These included pixel size, sampling angle, and energy peak optimization for $^{123}$I [16,17]. The main energy window was set at 20% (±10%) centered on 159 keV, while sub-energy windows for scatter estimation were set at 7% width on both the lower and upper sides of the 159 keV peak. The matrix size was 128 × 128, with a zoom factor of 1.23. Voxel size was 3.90 × 3.90 × 3.90 mm. The sampling angle was 5° per step, resulting in 72 views over a full 360° rotation. A non-circular orbit was used with step-and-shoot acquisition mode. Acquisition time per detector was varied at 10, 30, 50, and 100 s/view, and the total scan time was adjusted accordingly.

SPECT reconstruction was performed using iterative reconstruction (via 3D-OSEM), incorporating aperture collimation correction. The subset was fixed at 6; the product of the subset and iteration (SI product) was varied at 60, 120, and 180. The Gaussian filter was adjusted to 8, 10, and 12 mm. Scatter correction was applied using the triple energy window method, and attenuation correction was performed based on the CT images.

CT images for attenuation correction were acquired according to the vendor's recommendations, using a tube voltage of 130 kV, ref. mAs of 50 mAs (care dose type: AEC mean), a rotation time of 1.5 s, beam collimation of 6 x 2.0 mm, and pitch of 1. The images were reconstructed using the high-frequency smoothing function B08s, with a slice thickness of 4 mm, a reconstruction increment of 4.0 mm, and a field of view of 650 mm (voxel size: 1.27 × 1.27 × 4.0 mm).

Using the SPECT performance evaluation phantom JS-10, we evaluated dose linearity (DL), recovery coefficient (RC), scatter ratio (SR), and coefficient of variation (CV). For the NEMA IEC Body phantom, the SI product was fixed based on the JS-10 results. The contrast-to-noise ratio (CNR) and SUV were calculated while varying the acquisition time and Gaussian filter.

## Evaluation using the JS-10 phantom

The JS-10 phantom is a cylindrical phantom filled with a background activity of 1.73 kBq/mL, representing the mediastinum, as shown in Fig 1 [18]. Three evaluation discs were placed inside the phantom. Disc 1 was a cylindrical rod with a diameter of 30 mm, and its radioactivity concentration varied between two and 16 times that of the background. Disc 2 was a cylindrical rod with a fixed radioactivity concentration ten times higher than the background, while its diameter varied from 7 mm to 30 mm. Disc 3 was a rectangular water-filled prism placed on a uniform background. Based on previous studies using other radionuclides [14,15], we evaluated the following parameters. To minimize slice-selection bias when placing ROIs [19], we analyzed three slices per disc: the central slice and slices 10 mm above and below it.

DL was calculated as the correlation coefficient between the actual radioactivity measured in the rod and average SPECT count (R1−R4) for Disc 1.

For Disc 2, 30 mm circular ROIs (R5−R9) were placed on 7−30 mm diameter rods. The relative recovery coefficient ($RC_j$) for each rod (R6−R9) was calculated based on the maximum count of the 30 mm rod (R5). To evaluate the overall system resolution, total recovery coefficient ($RC_t$) was calculated from $RC_j$ using Equation (1):

$$RC_t = \sum (1 - RC_j)$$

(1)

In an ideal system, all $RC_j$ values would equal 1. However, due to partial volume effects, $RC_j$ in actual systems are typically underestimated. By integrating $1 - RC_j$ over the 7–30 mm diameter range in Equation (1), system resolution could be quantified. A higher $RC_t$ indicates lower resolution.

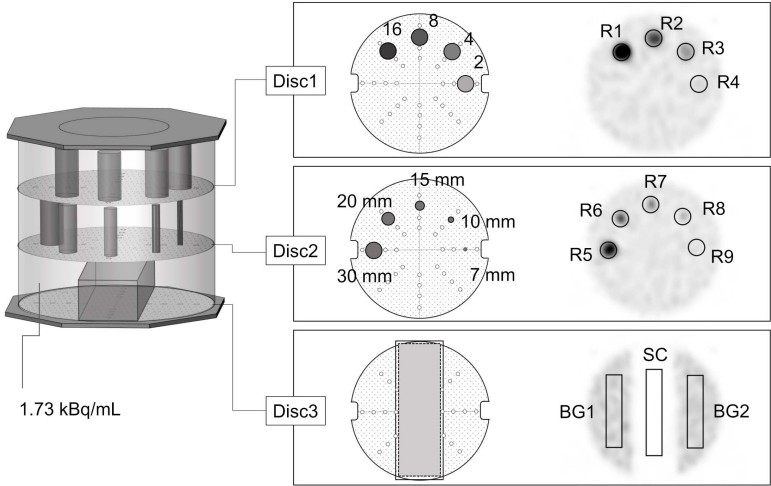

**Fig 1. JS-10 phantom.**

In Disc 3, rectangular ROIs (BG1, BG2, and SC) were placed in the background and water-filled region. The CV in the background and the SR were calculated using Equation (2):

$$SR = \frac{Average\ pixel\ value\ in\ SC}{Average\ pixel\ value\ in\ BG1\ and\ BG2} \times 100[\%]$$

(2)

### Evaluation using the NEMA IEC body phantom

For the NEMA IEC body phantom, the SI product was set to a single condition based on the optimal balance between image convergence and noise, as determined from the JS-10 phantom analysis. A hot sphere was placed inside the NEMA body phantom, and 1.42 kBq/mL and 15.81 kBq/mL of [123]I-MIBG were enclosed in the background and hot sphere, respectively, as shown in Fig 2. As shown on the right side of Fig 2, spherical voxels of interest were placed over the 17 mm hot sphere and background (n = 12) on the slice where the hot sphere exhibited the highest intensity. Contrast and noise were then calculated to evaluate the CNR [14]. In addition, spherical voxels of interest were placed over all spheres ranging from 10 mm to 37 mm in diameter; the maximum SUV (SUVmax) and mean SUV (SUVmean) were measured. Consistent with previous studies [19,20], the SUV was calculated by assuming a water density of 1 g/mL, thereby converting volume to an equivalent weight for normalization. The relative error rate for each acquisition time and hot sphere size was evaluated using the SUV obtained at 100 s/view as the reference.

### Statistical analysis

Continuous variables, including SPECT count data and calculated SUVs, were analyzed using descriptive statistics and are presented as mean ± standard deviation (SD).

For DL evaluation using the JS-10 phantom, Pearson's correlation coefficient was calculated to assess the relationship between the actual radioactivity concentrations in the rods and the average SPECT counts (R1–R4). A p-value of < 0.01 was considered statistically significant. All statistical analyses were conducted using BellCurve for Excel (Social Survey Research Information Co., Ltd., Tokyo, Japan).

## Results

### Evaluation results using the JS-10 phantom

The evaluation results for the JS-10 are presented in Table 1. Fig 3 shows the DL when the SI product was fixed at 120 and the Gaussian filter was changed; the DL when the Gaussian filter was fixed and the SI product was changed. At each acquisition time, the average SPECT count slightly decreased as the Gaussian filter increased. However, there was

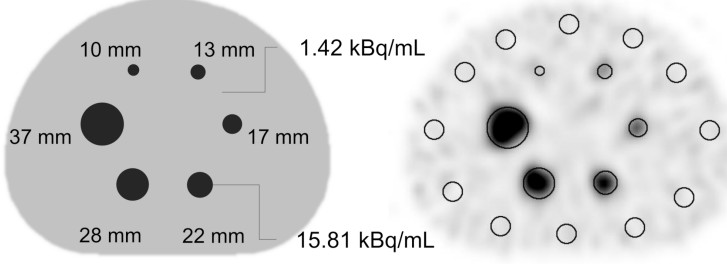

**Fig 2. NEMA IEC body phantom.**

**Table 1. Evaluation index results for the JS-10 phantom.**

| Gaussian filter [mm] | SI product | Time per view [s] | DL (r) | $RC_t$ | CV [%] | SR [%] |
|---|---|---|---|---|---|---|
| 8 | 60 | 10 | 0.999 | 2.25 | 37.42 | 10.07 |
| | | 30 | 0.998 | 2.12 | 24.52 | 8.97 |
| | | 50 | 0.998 | 2.39 | 20.40 | 5.80 |
| | | 100 | 0.998 | 2.46 | 13.66 | 2.67 |
| | 120 | 10 | 0.998 | 2.00 | 57.23 | 5.05 |
| | | 30 | 0.996 | 2.12 | 36.14 | 5.66 |
| | | 50 | 0.998 | 2.22 | 27.88 | 3.08 |
| | | 100 | 0.998 | 2.21 | 20.08 | 1.05 |
| | 180 | 10 | 0.999 | 2.09 | 66.45 | 8.69 |
| | | 30 | 0.998 | 2.11 | 34.57 | 7.78 |
| | | 50 | 0.998 | 1.94 | 35.82 | 3.72 |
| | | 100 | 0.998 | 2.11 | 24.70 | 0.55 |
| 10 | 60 | 10 | 0.999 | 2.38 | 29.67 | 10.10 |
| | | 30 | 0.998 | 2.42 | 18.75 | 7.56 |
| | | 50 | 0.998 | 2.48 | 16.23 | 5.88 |
| | | 100 | 0.998 | 2.57 | 10.87 | 2.75 |
| | 120 | 10 | 0.999 | 2.26 | 41.64 | 8.88 |
| | | 30 | 0.998 | 2.21 | 25.54 | 6.34 |
| | | 50 | 0.998 | 2.30 | 22.71 | 4.20 |
| | | 100 | 0.998 | 2.40 | 15.42 | 1.10 |
| | 180 | 10 | 0.999 | 2.25 | 48.97 | 8.67 |
| | | 30 | 0.998 | 2.10 | 30.02 | 6.17 |
| | | 50 | 0.998 | 2.21 | 26.87 | 3.76 |
| | | 100 | 0.998 | 2.31 | 18.55 | 0.59 |
| 12 | 60 | 10 | 0.999 | 2.47 | 23.94 | 10.14 |
| | | 30 | 0.998 | 2.48 | 15.13 | 7.71 |
| | | 50 | 0.998 | 2.57 | 13.08 | 5.95 |
| | | 100 | 0.998 | 2.64 | 8.78 | 2.84 |
| | 120 | 10 | 0.999 | 2.35 | 32.41 | 8.93 |
| | | 30 | 0.998 | 2.27 | 19.84 | 6.52 |
| | | 50 | 0.998 | 2.45 | 17.32 | 3.94 |
| | | 100 | 0.998 | 2.46 | 12.03 | 1.16 |
| | 180 | 10 | 0.999 | 2.38 | 37.14 | 8.72 |
| | | 30 | 0.998 | 2.35 | 22.74 | 6.40 |
| | | 50 | 0.998 | 2.49 | 17.32 | 3.87 |
| | | 100 | 0.998 | 2.52 | 14.16 | 0.63 |

Abbreviations: SI, subsets and iterations; DL, dose linearity; $RC_t$, recovery coefficient; CV, coefficient of variation; SR, scatter ratio

almost no change in the average count at each acquisition time when the SI product was changed. Pearson's correlation coefficient (r) was > 0.99 with a p-value <0.01 for all DL assessments, indicating a strong and statistically significant positive correlation. These findings confirm good linearity between measured radioactivity concentration and SPECT counts under all conditions.

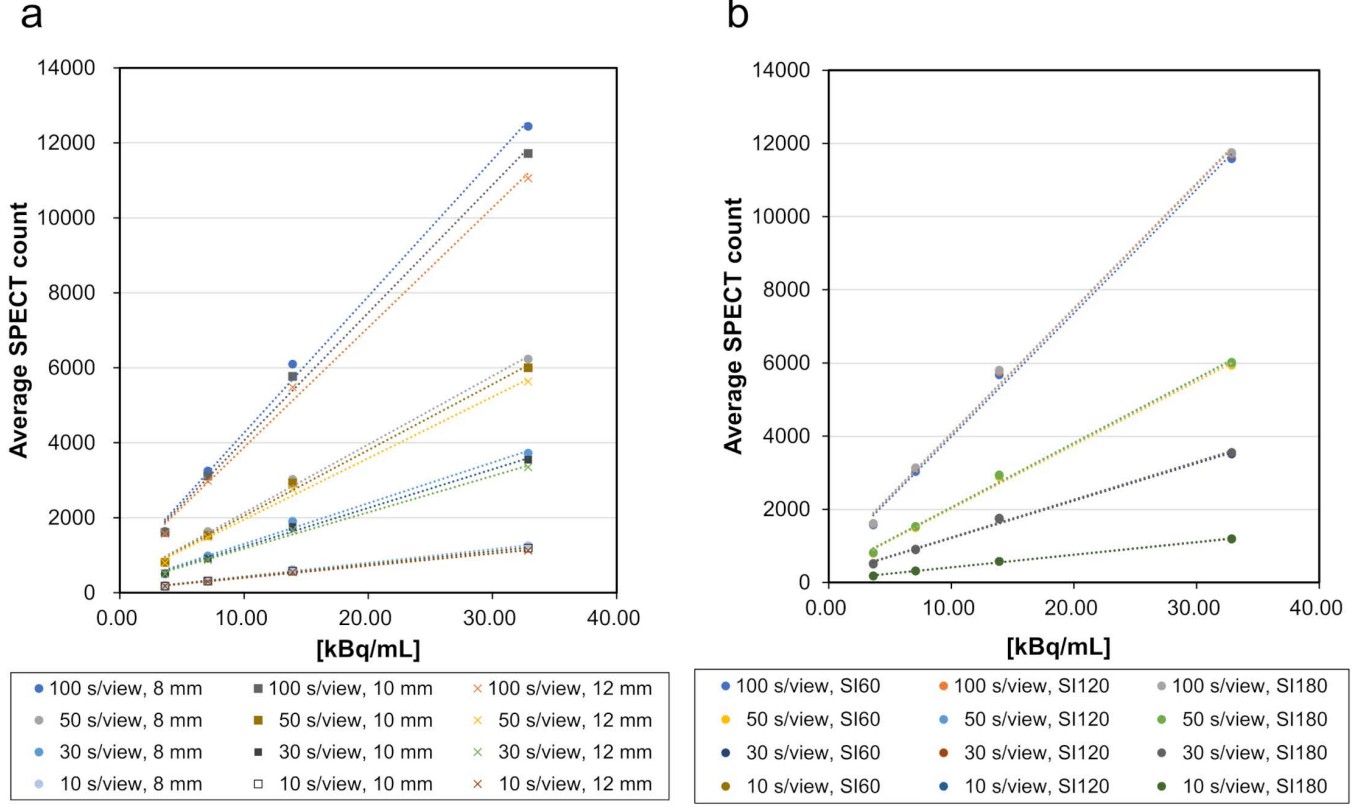

**Fig 3. Results of dose linearity.** This count-based assessment was performed to validate the proportional response of the system, a prerequisite for SUV quantification. Left: Relationship between the Gaussian filter and concentration linearity at each collection time (SI product fixed at 120). Right: Concentration linearity with a change in the SI product (Gaussian filter fixed at 10 mm). SI, subsets and iterations.

The RCs are shown in Fig 4. Comparing the results (Fig 4 and Table 1) for a given acquisition time, a larger full width at half maximum of the Gaussian filter or a smaller SI product generally resulted in lower $RC_j$ for individual rods, and consequently, higher $RC_t$.

The CV decreased with longer acquisition times. Notably, a substantial reduction in CV was observed when the acquisition time increased from 10 s/view to 30 s/view, whereas further increases to 50 s/view and 100 s/view resulted in comparatively smaller decreases. In contrast, the SR decreased as the acquisition time increased up to 100 s/view.

### Evaluation results using the NEMA IEC body phantom

Based on the spatial resolution and CV characteristics obtained from the JS-10 phantom, we fixed the SI product at 120 for evaluation with the NEMA body phantom. The contrast, BG noise, and CNR for the 17 mm sphere are shown in Fig 5. Contrast peaked at 30 s/view and showed minimal change beyond 50 s/view. However, BG noise decreased steadily as the time per view increased.

The SUVmax and SUVmean of each hot sphere under different Gaussian filter settings are presented in Fig 6. Using the 100 s/view acquisition as the reference standard, SUVmax for each sphere fluctuated as acquisition time decreased. This fluctuation lessened with the application of a wider Gaussian filter, though both SUVmax and SUVmean tended to decrease. Relative error rates, calculated using the SUV values at 100 s/view as the reference standard, were 25.86%

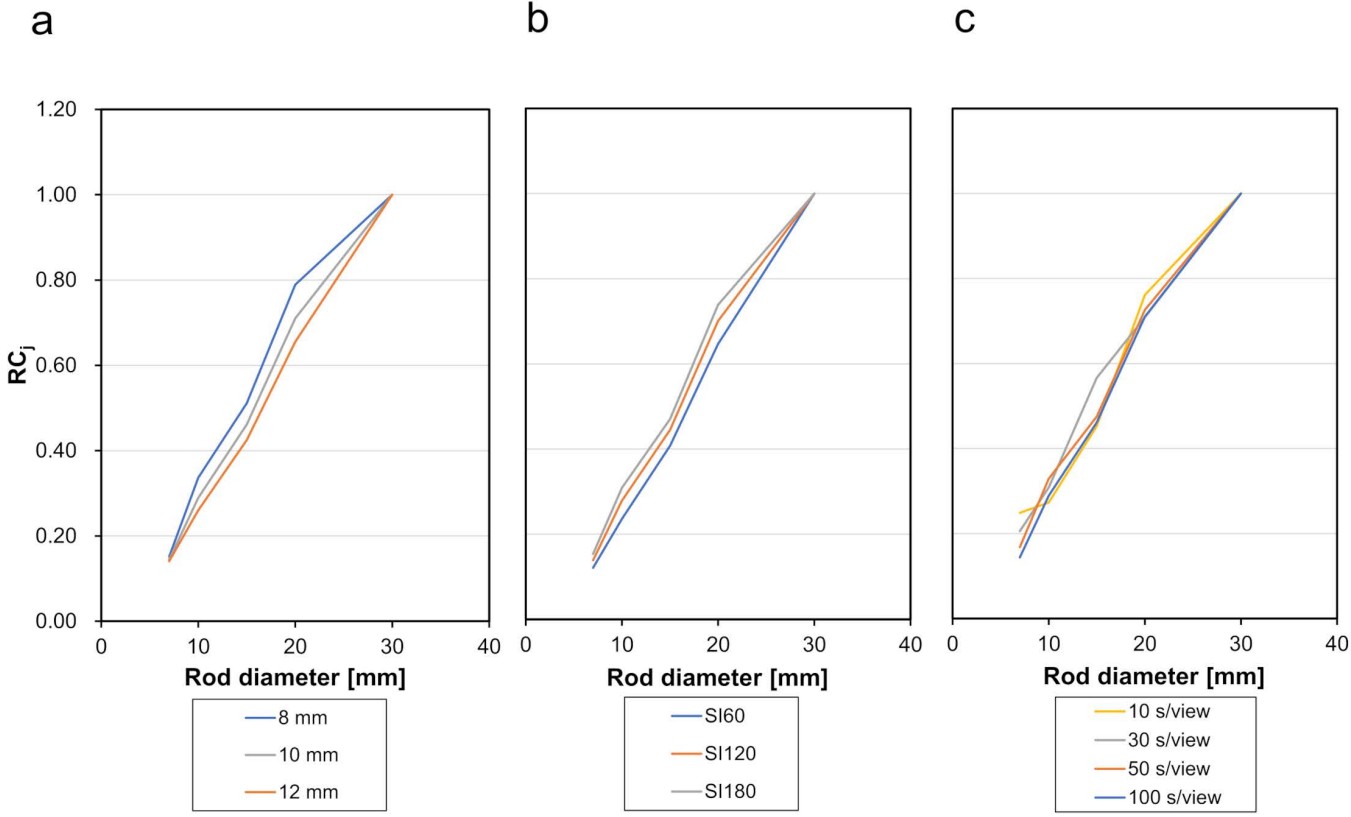

a b c

**Fig 4. Recovery coefficient.** The graphs show the relative recovery coefficient ($RC_j$) for individual rods of varying diameters. Left: Comparison of Gaussian filters when fixed at 100 s/view, an SI product of 120. Center: Comparison of SI products when fixed at 100 s/view with a Gaussian filter of 10 mm. Right: Comparison of acquisition times with a Gaussian filter of 10 mm and an SI product of 120. SI, subsets and iterations.

and 13.99% for SUVmax and SUVmean, respectively, at 10 s/view; 13.20% and 8.92% at 30 s/view; and 5.47% and 3.94% at 50 s/view.

The raw data used for the analyses are available as S1 Data.

## Discussion

Several clinical studies have utilized the SUV of [123]I-MIBG for cardiac assessment [20,21]. However, no foundational three-dimensional investigations have evaluated the presence or absence of myocardial uptake relative to the mediastinum. Furthermore, the reproducibility of quantitative indices remains limited due to substantial variability introduced by image noise. In this study, we employed both cylindrical and spherical phantoms to perform basic validation under imaging conditions more reflective of clinical settings, using realistic background activity levels.

### Evaluation using the JS-10 phantom

The evaluations using the JS-10 phantom served as a crucial initial phase to screen and optimize reconstruction parameters. Following the methodology of previous foundational studies [14,15], we utilized a series of SPECT-count-based physical indices—specifically DL, $RC_t$, SR, and CV—before proceeding to SUV quantification with the NEMA phantom. This stepwise approach allowed us to assess the intrinsic performance of the reconstruction algorithm under various conditions. For DL assessment, we evaluated the system's fundamental proportional response. The DL was excellent under

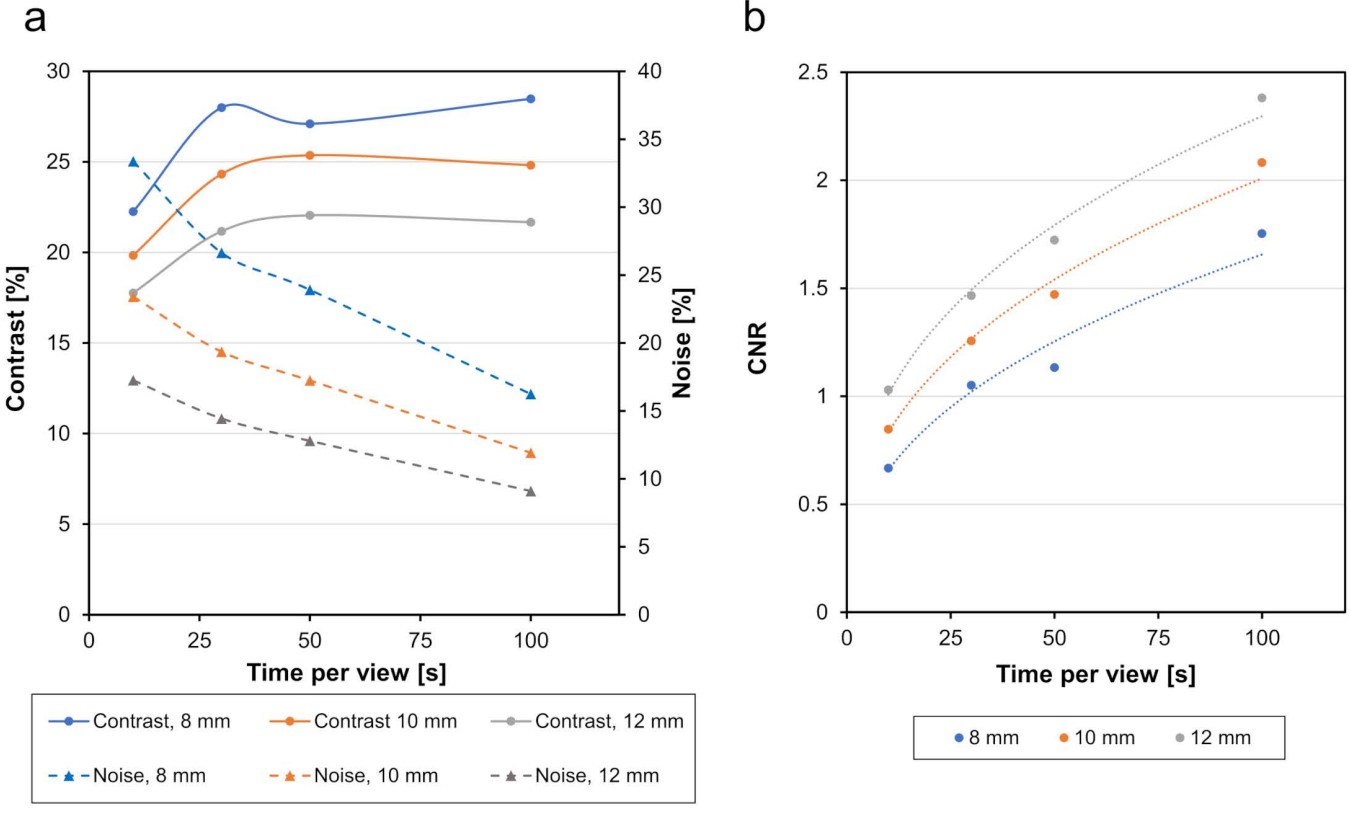

**Fig 5. Relationship between contrast, noise, and CNR.** CNR, contrast-to-noise ratio.

all conditions, with correlation coefficients ranging from 0.996 to 0.999. This indicates that, regardless of the acquisition time, reconstruction parameters, or the rod-to-background ratio (which ranged from 2:1–16:1), the 3D-OSEM algorithm accurately represented the relative radioactivity concentrations between high-uptake and background regions. These findings align with previous studies using technetium-99m [14,19], suggesting that a practical convergence point had been achieved for accurately reflecting the proportional relationship between high-uptake areas and background activity.

When the Gaussian filter and SI product were fixed, the SR decreased as the acquisition time per view increased, whereas the $RC_t$ generally increased. However, in some instances, $RC_t$ showed a reversal with longer acquisition times. For this $RC_t$ evaluation, we used a fixed 10:1 ratio to simulate a moderately reduced myocardial uptake, building upon our prior work that characterized the physics of RC dependency across various ratios [19]. When only the SI product was increased, both $RC_t$ and SR decreased. $RC_t$ serves as an index for evaluating high-uptake regions, while SR reflects background regions with no radiopharmaceutical accumulation. An accurate SR indirectly supports reliable evaluation of true defects, as cold region visibility is particularly important when using iterative reconstruction methods [14]. The convergence of $RC_t$ towards the true system resolution critically depends on applying a sufficiently high SI product, which in turn requires an adequate number of acquired counts [22,23]. The observed reversal in $RC_t$ at shorter acquisition times is likely due to the reduced number of photons contributing to image formation, resulting in a higher CV (i.e., greater statistical noise), which could adversely affect $RC_t$ measurements. However, it is important to note that simply increasing the SI product to improve $RC_t$ convergence can also lead to increased CV if not supported by sufficient count statistics. Furthermore, since SR decreased with both longer acquisition times and higher SI products, increasing the number of photons

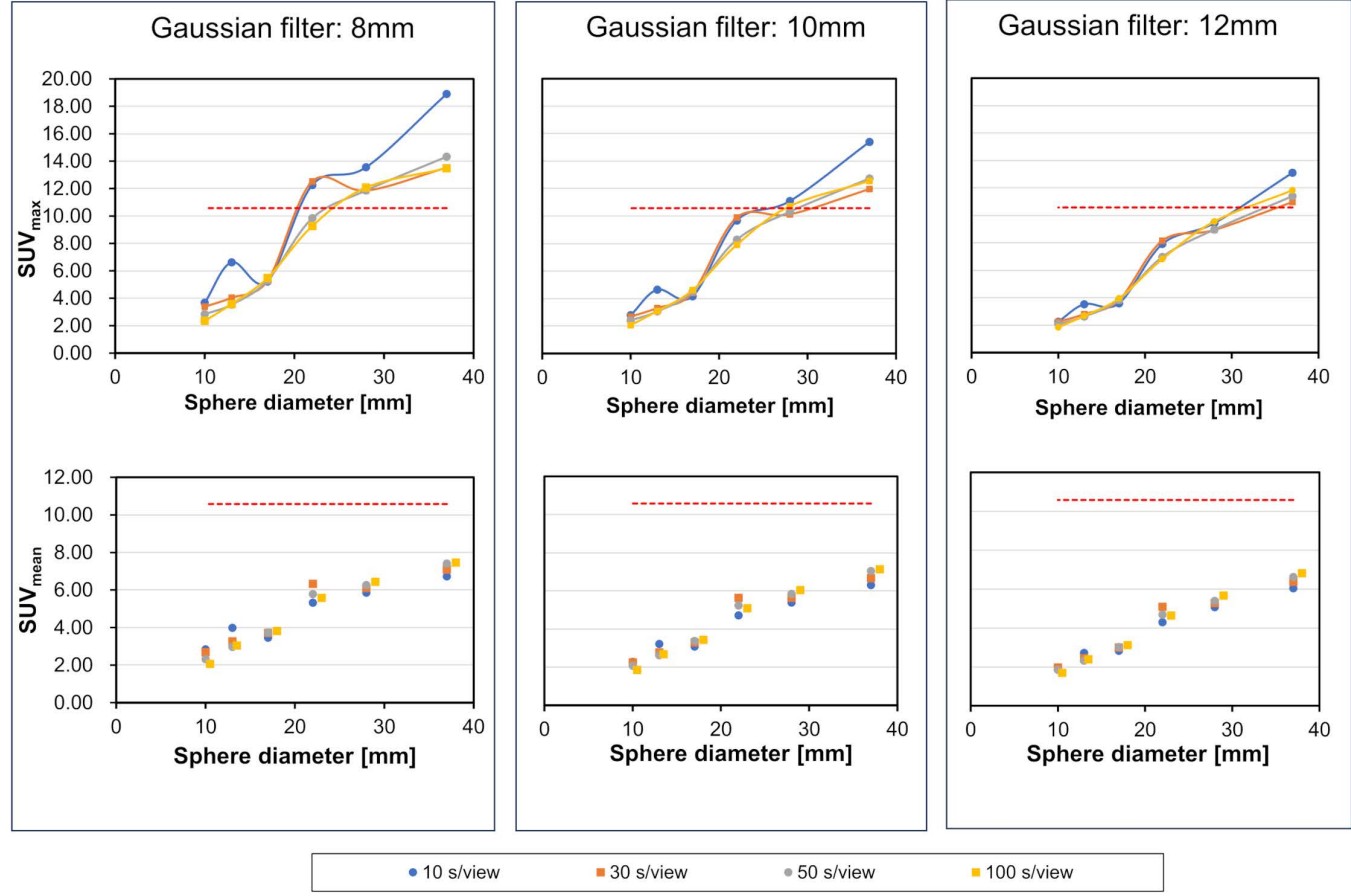

**Fig 6. SUVmax and SUVmean for each hot sphere when the Gaussian filter is changed.** (Left column) Gaussian filter: 8 mm; (Center column): 10 mm; (Right column): 12 mm. (Top row) SUVmax; (Bottom row) SUVmean. The red dashed line shows the theoretical SUV. SUV, standardized uptake value.

contributing to the image and maximizing the SI product within practical limits, is important [15] for correctly evaluating cold regions at the background level. When evaluating tracer accumulation under the assumption that the myocardium is located within the mediastinum, these results suggest that ≥50 s/view acquisition times provide stable evaluation.

Our SR and CV values were higher than those reported in previous studies using other radionuclides [14]. One contributing factor may be the limitations of the current triple energy window correction [24] in compensating for the high-energy gamma emissions of $^{123}$I. However, at 100 s/view acquisition time—where sufficient counts were obtained—the SR fell below 1% as the SI product increased, suggesting that this limitation could be mitigated by improving detector sensitivity. In addition, potential strategies to reduce CV include minimizing the SI product, increasing the Gaussian filter, and extending the acquisition time [20]. Notably, excessive emphasis on reducing CV may compromise resolution- and contrast-related indices such as $RC_t$ and SR.

### Evaluation using the NEMA IEC body phantom

As shown in Fig 5, contrast remained largely constant at ≥30 s/view when using an SI product of 120. The CNR increased as the noise decreased with longer acquisition times. Contrast is influenced by both the number of detected photons and the SI product applied to the projection data [19]. However, under the conditions used in this study, contrast peaks at

30 s/view for an SI product of 120. Depending on the extent to which resolution deterioration and noise suppression are required, extending the time per view and size of the Gaussian filter may be necessary.

As shown in Fig 6, the SUVmax for the hot sphere at 10 s/view and 30 s/view differed from the value measured at 100 s/view, likely due to the influence of noise. The error from the theoretical SUV value was also large. However, the relative error rate was approximately ≤5% when using an acquisition time of 50 s/view. The stabilization of SUV values at 50 s/view aligns well with the substantial improvement in CNR observed up to this point in Fig 5. This suggests that a sufficient signal-to-noise ratio is a prerequisite for achieving robust SUV quantification. Therefore, considering the reproducibility of analyses using both SUVmax and SUVmean, our findings indicate that an SI product of 120 and a Gaussian filter of 10 mm are optimal when acquiring data for 50 s/view or longer. A previous study on brain imaging recommended an SI product of 90 [25], whereas for myocardial imaging, Yasumoto et al. proposed an SI product of 100 [20], and Okuda et al. suggested a range of 90 − 120 [11]. This optimal SI product of 120 aligns with these previous reports. The broader range investigated in our study was crucial for demonstrating this optimum: an SI product of 60 yielded poorer quantitative accuracy due to under-convergence, while an SI product of 180 increased image noise without a corresponding benefit to SUV recovery, a particular concern for low-count $^{123}$I-MIBG imaging. However, those studies did not evaluate cold spots in background regions focused primarily on evaluating high-uptake areas and the appropriateness of the degree of accumulation.

The 128 × 128 matrix (3.90 mm pixel size) employed in this study offers higher spatial resolution than the 64 × 64 matrix recommended by the EANM guidelines for cardiac sympathetic imaging [26]. This choice was deliberately made to enhance the accuracy of quantitative SUV assessment—a primary objective of this research—by mitigating partial volume effects and enabling a more precise delineation of radiopharmaceutical distribution. Our findings suggest that even with smaller pixel sizes, appropriate optimization of acquisition time and 3D-OSEM reconstruction parameters can yield stable quantitative values. However, it is important to consider these parameter differences when applying these findings to routine clinical practice, which may follow different standard protocols.

## Limitations

This study has some limitations. First, it was conducted at a single facility using a single SPECT/CT device, and each imaging parameter set was evaluated with a single acquisition. This introduces potential bias related to acquisition conditions and image reconstruction algorithm used. Second, the optimization was based on a phantom simulating a standard-sized individual (60 kg) who received a full dose of 111 MBq of $^{123}$I-MIBG. The activity concentrations used reflect clinically plausible scenarios: the background (mediastinal) concentration (1.42–1.73 kBq/mL) accounted for approximately 20–30% early excretion of the administered MIBG and assumed uniform distribution of the remaining activity [18,20]. The hot sphere concentration (15.81 kBq/mL) was based on an estimated 1–2% of the effective in-body MIBG activity accumulating in the myocardium [27,28]. Third, our phantom setup does not account for the complex scatter and septal penetration effects originating from high-uptake organs adjacent to the heart, most notably the liver. This is a critical factor in clinical settings that can influence the quantification of nearby low-uptake regions such as the mediastinum. For these reasons, caution is warranted when generalizing our findings. The clinical application of these optimized parameters, particularly for mediastinal SUV, should be validated in settings involving different patient body sizes, administered doses, or through studies using more anthropomorphic phantoms. However, to enhance robustness and minimize inter-operator variability, we evaluated multiple cross-sectional slices and performed three-dimensional multi-point analyses using two distinct phantom types [14,19]. In addition, to date, no studies have assessed image quality or cold spot evaluation at BG-level accumulation using modern SPECT/CT.

## Summary and implications

In this study, we present the results of validation using two types of phantoms: a cylindrical phantom, which is less susceptible to inter-slice variation in quantitative evaluations, and a spherical phantom which accounts for three-dimensional partial

volume effects. Our study findings suggest that, for accurate evaluation of linearity in high-uptake regions, reliable representation of cold regions in the background and correction of radiopharmaceutical distribution in tissues surrounding the myocardium, image acquisition at ≥50 s/view is required. Optimal reconstruction conditions include an SI product of approximately 120 and a Gaussian filter of 10 − 12 mm. The clinical implication of these findings is significant for the advancement of quantitative MIBG SPECT. By establishing technical parameters that ensure the stable and accurate quantification of not only high-uptake myocardial regions but also low-uptake background areas like the mediastinum, our work provides an essential foundation for enhancing the robustness of 3D-based quantitative metrics, such as the myocardial-to-mediastinal SUV ratio. This, in turn, has the potential to support more reliable and reproducible cardiac assessments.

## Conclusion

We conducted a basic validation study using two types of phantoms to assess the accurate quantification of high myocardial uptake of [123]I. The ability to visualize cold spots in the background was largely dependent on the SI product and number of photons contributing to image formation (time per view). When evaluating the myocardium and surrounding tissues as background, an SI product of approximately 120 and a Gaussian filter of 10–12 mm were required for acquisitions of ≥50 s/view.

This work provides a foundational methodology for the robust quantification of [123]I-MIBG SPECT. The optimized parameters established herein are expected to support future clinical research aimed at improving diagnostic accuracy and evaluating treatment effects in a wide range of cardiac and neurodegenerative disorders.

## Supporting information

**S1 Data. The raw data used for the analyses in this study.**
(ZIP)

## Author contributions

**Conceptualization:** Masakazu Tsujimoto.

**Data curation:** Masakazu Tsujimoto, Masanori Watanabe, Kenta Nogami.

**Formal analysis:** Masakazu Tsujimoto, Masanori Watanabe.

**Funding acquisition:** Masayoshi Sarai.

**Investigation:** Masakazu Tsujimoto, Masanori Watanabe, Kenta Nogami, Hideki Kawai, Masayoshi Sarai.

**Methodology:** Masakazu Tsujimoto, Masanori Watanabe, Kenta Nogami.

**Project administration:** Masakazu Tsujimoto, Masayoshi Sarai.

**Resources:** Hideki Kawai.

**Software:** Masakazu Tsujimoto, Hideki Kawai.

**Validation:** Masakazu Tsujimoto.

**Visualization:** Masakazu Tsujimoto, Masanori Watanabe.

**Writing – original draft:** Masakazu Tsujimoto.

**Writing – review & editing:** Masakazu Tsujimoto, Masanori Watanabe, Masayoshi Sarai.

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
