## [Decision Letter · Decision Letter 0]

7 Nov 2024

Dear Dr. Tsujimoto,

We look forward to receiving your revised manuscript.

Kind regards,

Matteo Bauckneht

Academic Editor

PLOS ONE

Journal Requirements:

“The authors have no relevant financial or non-financial interests to disclose.”

Reviewers' comments:

Reviewer's Responses to Questions

**Comments to the Author**

1. Is the manuscript technically sound, and do the data support the conclusions?

Reviewer #1: Partly

Reviewer #2: Yes

2. Has the statistical analysis been performed appropriately and rigorously?

Reviewer #1: No

Reviewer #2: Yes

3. Have the authors made all data underlying the findings in their manuscript fully available?

Reviewer #1: Yes

Reviewer #2: Yes

4. Is the manuscript presented in an intelligible fashion and written in standard English?

Reviewer #1: Yes

Reviewer #2: Yes

Reviewer #1: Line 27: “single-photon emission computed tomography/computed tomography images were acquired by varying the acquisition time per detector” what does “per detector” mean, isn’t it rather the total acquisition time of the image volume?

Lines 40-42: “In conclusion, an acquisition time of ≥50 s/view with an SI product of approximately 120 and a Gaussian filter of 10−12 mm is desirable for evaluating the myocardium and surrounding tissues as background” It is the H/M ratio that allows us to judge the estimation of myocardial activity relative to the surrounding, this ratio was not considered in this study. This conclusion should be better specified.

Line 40. “For ≥50 s/view, the variations in maximum and mean SUVs decreased. The relative

standard error was <5%. » No explicit error analysis supports this claim, especially regarding mean SUVs.

Lines 68-70: « Since the basic verification of the comprehensive relationship between the myocardium and surrounding tissues has not been clarified, further research is required for clinical application », the manuscript raises an important issue that does not seem directly related to the study, as it does not provide any clarification on this issue

Lines 71-73: : » Therefore, in this study, we aimed to build evidence for clinical application and conduct a basic study on the convergence and quantification of images for the acquisition and reconstruction conditions for measuring SUV. »

How does this study provide clinical evidence? This is not the aim of the study, no clinical evaluation was conducted.

Line 133, equation 1 requires further explanation.

Lines 191-192: "The SUVmax and SUVmean results for each hot sphere when the Gaussian filter was changed are shown in Fig 6" This is not correct, Fig. 6 only shows the SUVmax.

Lines 193-195: "the relative error rates of the SUV at each time point were 25.86% and 13.99% for the SUVmax and SUVmean at 10 s/view, 13.20% and 195 8.92% at 30 s/view, and 5.47% and 3.94% at 50 s/view, respectively." These error rates cannot be deduced from Fig. 6. More precision is needed on the calculation of these error rates.

Reviewer #2: https://www.editorialmanager.com/pone/default2.aspx

In this work, Masakazu Tsujimoto et al investigated the relationship between image convergence and quantitation with the aim to establish the optimal measurement conditions for accurate SUV analyses of 123I-MIBG SPECT. The work deserves attention as 123I-MIBG SPECT is increasingly used for cardiac and brain examinations.

Comments:

-Line 59: The authors state that “does not eliminate the influence of overlap”. They should better precise what they mean here. Overlap between which structures?

-Line 63: The same holds true for this sentence. They should precise what they mean by “distribution of accumulation.”.

-Lines 82-83: more information about the NEMA IEC Body phantom should be provided, e.g. company, site, country and/or internet site.

-Introduction: The authors should specify why they chose these phantoms. Do they realistically represent organ structures? Why not using e.g. a four-dimensional NURBS-based cardiac-torso (NCAT) phantom, which provides a realistic model of the normal human anatomy and cardiac and respiratory motions?

-Lines 89-94: Please specify the reason for the choice of the settings of the main energy window, the sub-energy windows, the matrix size, the zoom, the sampling angle and the acquisition time per detector. Does the choice correspond to typical settings of clinical applications? How do the settings vary between examinations of the heart and the brain?

-Lines 101-108: The same question holds for the parameters of the CT acquisitions – are they routinely used and how they vary for the different body areas to examine?

-Discussion section: The authors should briefly discuss how patient body mass may affect the image quality and quantification accuracy of 123I-MIBG SPECT images. Would their conclusions hold for higher body mass?

-Although supporting data have been provided, they cannot be read. Probably they can only be read by the appropriate Siemens software.

Minor comments:

-Line 57: should be “… (ROI) in the myocardium …”

-Line 70: suggest rewriting to “... is required for routine clinical application of 123I-MIBG SPECT.”.

-Line 125: Suggest rewriting the caption of figure 1 to “JS-10 phantom”.

-Line 293: should be “Acknowledgements”.

-The unit in figure 2 is 15.81 kBq/ml.

-Throughout the manuscript, please change the unit of milliliters to ml (SI unit).

**Do you want your identity to be public for this peer review?** For information about this choice, including consent withdrawal, please see our Privacy Policy

Reviewer #1: **Yes: ** Boulanouar Abdel-Kader

Reviewer #2: No

---

## [Author Response · Author response to Decision Letter 1]

18 Nov 2024

Dear Editor and Reviewers,

We appreciate your review of our manuscript. In response to your comments, we have carefully revised our manuscript within the limitations of a Report Article. Our responses are provided below; the revisions are indicated in red in the revised manuscript. Deleted parts are indicated as strikethrough and in blue.

We have made minor adjustments to the graph's formatting to eliminate unnecessary white space.

Thank you again for your comments on our manuscript. We hope that the revised manuscript is now acceptable for publication as a Report Article.

Sincerely.

---

## [Decision Letter · Decision Letter 1]

23 May 2025

Dear Dr. Tsujimoto,

Thank you for submitting your manuscript to PLOS ONE. After careful consideration, we feel that it has merit but does not fully meet PLOS ONE’s publication criteria as it currently stands. Therefore, we invite you to submit a revised version of the manuscript that addresses the points raised during the review process.

We look forward to receiving your revised manuscript.

Kind regards,

Hugh Cowley

Staff Editor

PLOS ONE

Reviewers' comments:

Reviewer's Responses to Questions

**Comments to the Author**

Reviewer #1: All comments have been addressed

Reviewer #2: All comments have been addressed

Reviewer #3: All comments have been addressed

Reviewer #4: (No Response)

2. Is the manuscript technically sound, and do the data support the conclusions?

Reviewer #1: Yes

Reviewer #2: Yes

Reviewer #3: Partly

Reviewer #4: No

3. Has the statistical analysis been performed appropriately and rigorously?

Reviewer #1: Yes

Reviewer #2: Yes

Reviewer #3: No

Reviewer #4: No

4. Have the authors made all data underlying the findings in their manuscript fully available?

Reviewer #1: Yes

Reviewer #2: Yes

Reviewer #3: Yes

Reviewer #4: No

5. Is the manuscript presented in an intelligible fashion and written in standard English?

Reviewer #1: Yes

Reviewer #2: Yes

Reviewer #3: No

Reviewer #4: No

Reviewer #1: All the comments have been properly addressed. The answers are acceptable and correctly formulated.

Reviewer #2: The authors addressed well all my previous comments, thanks. They also pointed out the limitations associated to their study.

Reviewer #3: You determined the optimal acquisition time and reconstruction parameters for myocardial MIBG using the performance evaluation phantom. In particular, he emphasizes the novelty of considering low-accumulation areas such as the mediastinal region. However, the study has several major faults.

First, only the hot rod region was evaluated in the NEMA and JS-10 phantoms (Disc 2). It is important to evaluate the defect in the myocardial MIBG phantom. Hence the need for a cold rod evaluation would be necessary. In particular, the JS-10 phantom can be verified similar to the MIBG defect evaluation by incorporating the radioactivity concentration of the myocardium in the BG and the radioactivity concentration of the mediastinum in the Rod. This evaluation should be added.

Second, this study does not simulate the human body. In particular, MIBG has a high accumulation in the liver. The study does not consider the effect of scattered rays from the liver. You mention the limitations of the scatter correction in your discussion as well. Therefore, it is uncertainty whether this phantom result can be clinically useful. The fact that the scattered rays from the liver affect especially in the mediastinum must also be considered in your novelty. At least a clinical evaluation should be added to show the validity of this condition.

Finally, your English text is poor and contains many errors, including multiple definitions of abbreviations. If you wish to resubmit your paper, we strongly recommend that you proofread it in English.

Other points of concern are commented below.

Line 58

Add the full spelling of H/M

Line 103-105

You have determined the SPECT parameters based on Reference 16. However, its guidelines are standard protocols by MPI. SPECT parameters by Tc and Tl differ from those by I-123 in terms of dose, accumulation rate, and collection counts. A standard protocol for MIBG has already been proposed in the EANM. If your protocol differs from that standard protocol, you should either re-experiment or consider the impact of the differences in discussion.

In particular, matrix and pixel size are significantly different.

Line 112-113

You set the SI product to 60, 120, and 180. The evidence for this set up should be provided. The 3D-OSEM with Flash3D algorithm has a proposed SI product of 90-120 in MPI. However, MIBG imaging is not sure if it is the same protocol due to low acquisition counts. In particular, the SI product of OSEM affects spatial resolution and image uniformity, and is also involved in the accuracy of SUVs. The relevance of the SI product should be mentioned in your paper.

Line 117-123

You indicated two different CT reconstruction parameters. What does this mean? Do you evaluate the effect of attenuation correction due to differences in CT imaging conditions? You should add details on how you utilized these two parameters. If there is only one type of reconstruction protocol used as attenuation correction, one or the other should be removed as they are not relevant to the paper.

Line 135-136

You set the radioactivity concentration ratio of Rod to BG to 10. The reasons for this setup are important. You also mention in your objectives the accuracy of quantitation in low-accumulation cases. Please provide additional examination of the effect of different radioactivity concentration ratios on count recovery. Or discuss the influence of count recovery in your discussion with reference to previous studies.

Line 139

The full spelling of ROI has already been described in Line 59.

Line 145-146

You have evaluated the correlation between the SPECT counts and the actual radioactivity concentration. Is the actual radioactivity concentration image-based or dose calibrated? Since this study is based on quantitation, the DL should be calculated between Bq/mL of dose calibrator and Bq/mL of image-based Bq/mL

Line 147-149

You are calculating the relative recovery factor, but the 30mm diameter rods are already affected by the partial volume effect. It should either be an absolute recovery factor or a relative recovery factor with the outer container without Disc containing the same radioactivity concentration as Hotrod as a reference. The relative recovery coefficient does not reveal the percentage of each Rod that underestimates the quantitative value.

Line 163-164

How do you consider this relationship of radioactivity concentration in relation to your research objectives? Some reports on MIBG SUVs have already been published by Sito S et al as follows. Your determination of radioactivity levels should be set up with clinical feedback. In particular, the radioactivity concentration is important in relationship to the spill-in and spill-out effects.

Saito S, et al. Absolute quantitation of sympathetic nerve activity using [123I] metaiodobenzylguanidine SPECT-CT in neurology. EJNMMI Rep. 2024; 8(1): 15.

Saito S, et al. Three-Dimensional Heart Segmentation and Absolute Quantitation of Cardiac 123I-metaiodobenzylguanidine Sympathetic Imaging Using SPECT/CT. Ann Nucl Cardiol.

2023; 9(1): 61-67.

Line 177 Table 1

Correlation coefficients are calculated in Table 1, but there is no description of the statistical method. Which is Pearson correlation coefficient or Spearman('s) rank-correlation coefficient? It is necessary to show whether there is a significant relationship for all correlations.

Line 196

the larger the RCt were ?

Line 203-204

I don't understand if it's an “decrease was gradual” or not due to four measurement points only.

Line 208-209

While it is important to consider spatial resolution and CV, it is also important to consider whether the point of convergence of the 3D-OSEM method calculation has been achieved.　However, you have not shown the convergence point of the 3D-OSEM method. Please indicate if this is also a reasonable point of convergence for successive iterative reconstruction.

Line 213-216

You show the SUVs in each sphere for each collection time. However, you have not described the results.

Figure 3

The linearity between the dose calibrator and the Bq/mL on the SPECT image is important, as mentioned above.

Line 220-222

There is no explanation for a, b, and c in the figure. In addition, what does the 100s/view error bar mean? Is it a relative error? We think the error bars in this figure are inappropriate. If it is common practice to show error bars for relative errors in the figure, please provide a rationale for this.

Line 226

Please include references.

Line 230-232

Background level is important, but the relationship between cardiac accumulation and background is also important in quantitative evaluation. You have failed to evaluate that point in detail.

Line 236-239

DL is only linearity, and it is inappropriate to evaluate the convergence of image reconstruction parameters.

Line 249-250

To corroborate the abrupt noise, you should add the resulting SUV variation in the sphere.

Line 284-285

You emphasize that the evaluation of background equivalent to the mediastinum is an advantage over other studies. However, its clinical significance has not been described. Hence, the importance of optimizing image reconstruction processing conditions at the BG level has not been demonstrated. You should discuss the clinical significance of this study.

Line 290-292

The evidence of radioactivity concentration is insufficient. You have determined the radioactivity concentration from previous study. How much of this do you assume accumulated in the myocardium and blood pool in patients who received MIBG? It is not possible to determine whether the radioactivity concentration settings in this study are related to the accumulation of myocardium and blood pools by the clinical cases. Your study especially emphasizes the importance of optimizing Bq/mL in the mediastinum. The radioactivity concentration in the mediastinal part of the MIBG should be measured in several cases and the mean and SD should be added.

Line 303

We have not been able to evaluate your results regarding the visualization of defects. Also, Disc 3 is inappropriate as a defect evaluation.

If a defect evaluation is to be performed, a cold rod evaluation should be performed in Disc 1 and Disc 2.

Line 314

The acquisition time is also important, but since the results are only for this phantom, it would be easier for the reader to understand if the myocardial and mediastinal counts were shown as Projection counts.

Reviewer #4: This study has multiple major limitations. I have listed my concernes below,

1- Line 62 – Blood pool can also overlap with the myocardium.

2- Line 91 – Lower values indicate better resolution, so the term “poorer” should be used instead of “lower.” Also, specify that this refers to spatial resolution for clarity.

3- Line 93–94 – There are multiple poorly written sentences, where language need to be improved. These affect the overall clarity. I would suggest to the authors to have their paper proofread by an English native speaker. One example is “fluctuation of three-dimensional ordered subset expectation maximization (3D-OSEM) reconstruction of factors such as counts and reconstruction parameters.” What type of parameters are meant and unclear how counts impact reconstruction? Is image quality meant here or quantitation?

4- Line 105 –The photon energy for I-123 is 159 keV, not 158 keV.

5- Line 110 – The paper should specify the number of views around the 360-degree rotation, total scan time, and whether the acquisition was performed in continuous mode or step-and-shoot mode.

6- The paper does not describe how the system was calibrated to convert counts into activity concentration (Bq/ml). This is essential for understanding the reliability of the SUV values. It’s unclear (and unusual) why SUV values were evaluated with phantoms rather than focusing on quantitative accuracy. Normalizing against weight for SUV does not make sense for phantoms.

7- The paper does not include repeated measurements to establish statistical significance of the results shown. The findings of this study thus remains of unclear significance.

8 - The paper should explain how the phantom was filled with activity and if the concentrations were based on realistic levels. Were these concentrations clinically realistic? Were the total counts consistent with clinical practice?

9 - SUVmean in Figure 6 has very high error bars, which is unusual since SUVmean is typically stable over noise (average over a large area). The reason for this variability should be clarified. Also, the other schemes shown in Figure 6 do not have any error bars, making it difficult to interpret or compare the results. Including consistent error bars across all schemes and for all the metrics are essential.

**Do you want your identity to be public for this peer review?** For information about this choice, including consent withdrawal, please see our Privacy Policy

Reviewer #1: **Yes: ** Abdelkader Boulanouar

Reviewer #2: **Yes: ** Nicolau Beckmann

Reviewer #3: No

Reviewer #4: No

---

## [Author Response · Author response to Decision Letter 2]

26 Jun 2025

We thank the Editor and Reviewers for their constructive feedback. We have addressed all comments and have provided a detailed, point-by-point response in the attached file, "Response to Reviewers." The manuscript has been revised accordingly, and we hope it is now suitable for publication.

---

## [Decision Letter · Decision Letter 2]

5 Aug 2025

Optimization of acquisition time and reconstruction parameters for quantitative single-photon emission computed tomography/computed tomography using iodine-123 metaiodobenzylguanidine

PONE-D-24-40275R2

Dear Dr. Tsujimoto,

We’re pleased to inform you that your manuscript has been judged scientifically suitable for publication and will be formally accepted for publication once it meets all outstanding technical requirements.

Kind regards,

Lorenzo Faggioni, M.D., Ph.D.

Academic Editor

PLOS ONE

Reviewers' comments:

Reviewer's Responses to Questions

**Comments to the Author**

Reviewer #1: (No Response)

Reviewer #2: All comments have been addressed

2. Is the manuscript technically sound, and do the data support the conclusions?

Reviewer #1: (No Response)

Reviewer #2: Yes

3. Has the statistical analysis been performed appropriately and rigorously?

Reviewer #1: (No Response)

Reviewer #2: Yes

4. Have the authors made all data underlying the findings in their manuscript fully available?

Reviewer #1: (No Response)

Reviewer #2: Yes

5. Is the manuscript presented in an intelligible fashion and written in standard English?

Reviewer #1: (No Response)

Reviewer #2: Yes

Reviewer #1: (No Response)

Reviewer #2: All comments have been well addressed by the authors, thanks. I recommend publication of the manuscript in its present version.

**Do you want your identity to be public for this peer review?** For information about this choice, including consent withdrawal, please see our Privacy Policy

Reviewer #1: **Yes: ** Abdel-Kader Boulanouar

Reviewer #2: **Yes: ** Nicolau Beckmann

---

## [Editor Report · Acceptance letter]

PONE-D-24-40275R2

PLOS ONE

Dear Dr. Tsujimoto,

I'm pleased to inform you that your manuscript has been deemed suitable for publication in PLOS ONE. Congratulations! Your manuscript is now being handed over to our production team.

Kind regards,

on behalf of

Dr. Lorenzo Faggioni

Academic Editor

PLOS ONE